# Exploring the Relationships between Multilevel Built Environments and Commute Durations in Dual-Earner Households: Does Gender Matter?

**DOI:** 10.3390/ijerph20064851

**Published:** 2023-03-09

**Authors:** Xiaoquan Wang, Weifeng Wang, Chaoying Yin

**Affiliations:** 1College of Civil and Transportation Engineering, Hohai University, Nanjing 210098, China; 2College of Automobile and Traffic Engineering, Nanjing Forestry University, Nanjing 210037, China

**Keywords:** built environment, commute duration, household couples, difference, spatial levels

## Abstract

The links between built environments (BE) and commute durations have been extensively studied. However, relatively few studies have considered the effects of BEs at different spatial levels within a unified framework, or identified the gendered relationships between BEs and commute durations. Using survey data from 3209 household couples in 97 Chinese cities, this study investigates the effects of neighborhood- and city-level BEs on commute durations and the potential differences in these effects between the male and female members of the same household couple. A multi-group generalized multilevel structural equation model is applied to reveal the gendered relationships between neighborhood- and city-level BEs and commute durations. The findings suggest that the BE variables at two levels have significant effects on the commute duration. The mediating roles that the traffic congestion, car ownership, and commuting modes play in linking these BEs and commute durations are confirmed. Both levels of the BE variables are more influential factors for males’ commuting durations. These findings have policy implications for the design of gender-equal transportation systems.

## 1. Introduction

Car-oriented suburbanization and urban sprawl have increased traffic congestion and commute durations. Almost all commuters in large cities over the world are suffering from commuting burdens. For example, the number of inter-city commuters in Germany has increased from 2,442,630 in 2004 to 3,123,924 in 2014 [1], which has caused increased traffic loads and frequent congestion [2]. In China, the average one-way commute duration was more than 36 min in 42 major cities in 2020. Moreover, more than 24% of the commuters in these cities had an average one-way commute duration that was longer than 45 min. In Beijing, this proportion was up to 43%. Commute durations need to be minimized as commuting is a stressful activity. Long commutes not only waste time but also lower people’s quality of life. Thus, reducing traffic congestion and commute durations has become an essential goal in the field of urban planning.

To reduce commute durations, many transport practitioners and researchers have studied their relationship with the built environment (BE). Most of them focus on Western cities [3], which are characterized by a long history of industrial and infrastructure developments. However, evidence from Western cities may not be generalizable to non-Western countries (e.g., China), owing to different urban development levels [4]. It is estimated that 350 million people will move from rural to urban areas by 2025 during the urbanization process in China [5]. Rapid urbanization not only causes greater car dependence and traffic congestion, but also leads to urban sprawl and changes in the BE [6]. These characteristics make China a good case study for understanding the link between BEs and commute durations.

This study identifies the effects of both neighborhood- and city-level BE variables on commute duration, with a focus on gender differences, using survey data from 3209 household couples in 97 Chinese cities. This study contributes to the international literature on BEs and commute durations by answering the following questions: (1) Do the two levels of BE variables correlate with commute durations? (2) Do the effects of the BE variables differ between the male and female members of the same household couple?

The reminder of this paper is organized as follows. The next section reviews the existing studies related to BEs, commute durations, and gender differences. The data, conceptual framework, and method used in this study are then introduced. Next, the estimation results are presented. Finally, the key findings are discussed, followed by the conclusions.

## 2. Literature Review

### 2.1. BE and Commute Duration

To reduce the negative effects of long commutes, previous studies have explored the relationship between BEs and commute durations [5,7,8,9,10]. The association between the neighborhood-level BE and commute durations has been widely explored. Researchers tend to use the 5Ds (i.e., density, diversity, design, distance to transit, and demand management) to measure the BE in the previous studies on BEs and travel behavior [11]. The 5Ds have been also extensively studied in the context of BEs and commute durations [8]. Although no concrete consensus has been reached on the relationship between Bes and commute durations, owing to the variations in research contexts and methods in previous studies, some strategies, such as promoting high-density developing strategies and improving destination accessibility, are widely considered to be effective for reducing commute durations. For example, Van Acker and Witlox found that land use mix and distance to transit at the traffic analysis zone level were important factors of commute durations [12]. Antipova et al., used a multilevel regression model to explore the impact of urban land use on commute durations [8]. The results suggest that the residential density and land-use types in neighborhoods have significant impacts on commute durations. Using survey data from Beijing, Zhao found that the neighborhood-level residential density and public transit accessibility were determinants of commute durations for both active and non-active commuters [9].

Apart from the neighborhood-level BE and commute durations, some prior studies have examined the BE beyond residential locations [5,13,14,15]. For example, Mouratidis found that compact-city residents were more likely to have shorter commutes based on survey data from Norway [15]. Unlike in urban areas, not all people in suburban locations have longer commutes [16]. By analyzing the effectiveness of polycentric development patterns, Jun found that inner-city subcenter residents experienced decreases in their commute durations, whereas many workers relocated their households to new suburban subcenters, leading to longer commutes [17]. The BE variables at workplace locations and around the route areas also affect commute durations. Although there exist arguments about the effect sizes of the BE at residences and beyond residential locations (e.g., workplace locations), prior studies have demonstrated that the BE variables at workplace locations affect various commute-related terms, such as modes [18], distances [19], vehicle miles traveled [20], and well-being [21]. In addition, the BE variables along a route usually trigger various kinds of travel behavior, and their relationships have been confirmed by many prior studies [22,23]. As the city-level BE generally reflects the characteristics of the BE beyond residential locations (e.g., workplaces and route areas), it is critical to consider the BE at the neighborhood and city levels within a unified framework. Although some recent studies have confirmed that both levels of the BE variables are determinants of travel-related parameters [4,20], prior studies focusing on commute durations have tended to examine the impacts of the BE variables at different levels, separately. Considering the BE at both the neighborhood and city levels within a unified framework can provide further refined implications and avoid biased results, owing to omitting essential variables.

### 2.2. Gender Differences in Commuting Travel

Although interdependencies exist between household members’ activities and travels, household couples tend to have different commute behaviors, owing to gendered resources, family roles, work disparities, and perception differences [24,25,26,27]. Many prior studies have demonstrated the gender differences in commute-related outcomes, with a primary focus on commute distances or time [25,28,29,30]. Dual-earner couples tend to strike tradeoffs among residential locations, workplace locations, commute costs, and wage rates, and these tradeoffs are constrained by social roles, labor market segmentation, and spatial environments [24]. Thus, the gender gaps in commuting-related issues may be influenced by multifaceted factors. For instance, Chidambaram and Scheiner found that male partners commuted longer than female partners do, and economic prospects, car access, and time spent on unpaid work can increase or decrease the gender gap [25]. Based on the National Household Travel Survey in the United States, Hu also found gender gaps in commuting distances [24]. Additionally, household structures and race have interactive effects on these gender gaps. Apart from distance and time, gender differences also exist in travel modes. Many prior studies have indicated that women are less likely to use cars [27,31]. Although some studies show that the trend has diminished over time, gender differences in attitudes toward car travel persist [24].

However, relatively few studies have provided evidence for the gender differences in the association between BEs and travel behavior. The male and female members of the same household couple not only attach a different importance to different life domains, but also have different perceptions of spatial attributes [28]. For instance, Yang et al., found that the BE affected the car use of all the household members except for the household head, based on survey data from Nanjing [32]. Using longitudinal survey data from China, Wang et al., explored the effects of the BE on travel mode switching [27]. The results show that BE changes have a greater impact on men’s travel mode switching behavior. However, it is unknown whether and how the BE variables have different impacts on the commute durations between the male and female members of the same household couple.

### 2.3. Transport Inequity in China

China has experienced impressive economic growth since implementing market-oriented reforms. This growth causes widening income gaps and gives rises to social inequality in post-reform China [33]. In this context, transportation poverty and inequality in Chinese cities have received increasing research attention. The commuting patterns in Chinese cities are not comparable to those in Western countries or other parts of Asia. In recent decades, most Chinese cities have witnessed an enormous increase in urban expansion and motorization. Priority is given to motor vehicles in road construction within Chinese cities [6], creating divisions among different groups of residents. Low-income workers tend to have limited transport options and disadvantageous housing locations [34]. Owing to the complex urban structures and transport systems in China, daily commutes and non-commute travels in China are influenced by many factors [35], which are also closely related to the causes of transport poverty. These factors include a lack of access to cars, uneven access to alternative transport options, inadequate public transport provision, job-housing imbalance, and institutional factors [33]. Table 1 summarizes some of the results of prior studies that have examined the impacts of the above factors.

### 2.4. Comments on Previous Studies

Based on this brief review, two main gaps are identified in the literature. First, most prior research on commute durations focuses on the neighborhood-level BE. The impacts of neighborhood- and city-level BEs on commute durations have been poorly investigated within a unified framework. Some researchers have investigated the impacts of two-level BEs on travel behavior and found that the city-level BE plays a more important role [52,53]; however, few cross-scale studies have focused on commute durations. Exploring the impacts of both levels of the BE on commute durations is necessary, owing to the following reasons. On the one hand, the spatial uncertainty of the impacts of the BE variables on travel behavior suggests that considering only the residential BE variables may produce biased results, owing to the modifiable areal unit problem [54]. On the other hand, workplaces are usually located beyond residential neighborhoods [6]. Thus, it is necessary to include the BE at different spatial levels, in order to better understand the impact of the BE variables on commute durations. Because the city-level BE variables generally reflect the features of residences, workplaces, and surrounding routes, it is necessary to include the effects of the neighborhood- and city-level BE simultaneously.

Second, most prior studies have used an individual as the analysis unit when examining the link between BEs and travel behavior, and ignore the gender differences between the male and female members of the same household couple. Although some researchers have found spousal similarity in a multidisciplinary behavior analysis [27,55,56], the impacts of factors on travel behavior may differ between the male and female members of the same household couple, because they usually attach various importance to the BE and life domains [32]. To design gender-equal transportation systems, it is necessary to understand whether and how the impacts of the city- and neighborhood-level BE variables differ between the male and female members of the same household couple.

Based on existing empirical evidence, this study aims to investigate the impacts of the BE at different spatial levels on commuting durations and how these impacts differ between the male and female members of the same household couple. To achieve these objectives, we hypothesize that both the neighborhood- and city-level BE variables affect commute durations in this study. It is recognized that the impacts of the BE variables on commute durations are hierarchical. Moreover, we hypothesize that the impacts of the BE variables on commute durations differ between the male and female members of the same household couple. The intermediary nature of traffic congestion, car ownership, and commuting modes is also considered in this study.

## 3. Methodology

### 3.1. Data Sources

The data for this analysis are obtained from the 2014 China Labor-force Dynamics Survey (CLDS), which provides a large-scale dataset to examine the influential factors of commute durations. The survey is conducted by Sun Yat-sen University and it covers 29 provinces in China. Owing to the huge differences in the population sizes and economic development levels of Chinese cities, a multi-stage cluster and stratified PPS sampling method is used to guarantee the representativeness of the national labor force [57]. First, the selected provinces are divided into nine categories based on their locations and population sizes. Cities are then randomly selected within each category according to their gross domestic product (GDP) rankings. The number of cities in each category is determined according to the proportion of its labor force within the national labor force. In each city, neighborhoods are randomly selected based on their GDP rankings and proportion of the migrant population. Finally, households are randomly selected from each neighborhood. The CLDS provides information about individuals, households, and communities. Except for those from the city-level BE, all the variables are directly extracted from the dataset or indirectly calculated based on the information in the dataset. Because we focus on dual-earner households in this study, the individuals from households with other types of household structures are removed. After removing missing information, the survey data of 3209 household couples from 329 communities and 97 cities are used for this analysis.

The dependent variable is one-way commute duration, which is directly extracted from the dataset. Individual socio-economic variables are also recorded in the dataset. The statistical descriptions of the variables at the individual level are presented in Table 2.

Among the neighborhood-level BE variables, the distance to transit stations and the distance to the CBD are directly extracted from the dataset. The population sizes and areas of the surveyed neighborhoods are recorded in the dataset and used to calculate the population density. The facility diversity is measured using the sum of seven types of facilities (i.e., sports facilities, reading rooms, elderly activity rooms, squares, playgrounds, hospitals, and banks). Thus, the value of the facility diversity is between 0 and 7. A higher value indicates higher diversities.

The city-level variables include traffic congestion and the city-level BE variables. Traffic congestion is treated as a binary variable, according to whether the city entered the top ten congested cities at least once in China in 2014. These rankings are released by AMAP every quarter, which is one of the most popular web mapping service applications in China. Chinese cities are ranked according to their congestion delay indexes during peak hours. The congestion delay index is obtained by dividing actual travel time by free-flow travel time. In total, two datasets are used to calculate the city-level BE variables. First, the land use mix and bus coverage rate are extracted from Beijing City Lab (see https://www.beijingcitylab.com/ for details accessed on 1 February 2023). The land use mix is calculated based on eight categories of POIs (points of interest), including commercial sites, office buildings, transport facilities, governments, educational facilities, residential communities, green spaces, and others. The bus coverage rate is also directly extracted from the dataset. Second, as the CLDS provides the city information, the population density, metro, public transit supply, and road areas per person are directly extracted or indirectly calculated based on the data from the China City Statistical Yearbook 2014. In this study, polycentricity refers to the degree to which employments or inhabitants are concentrated around sub-centers. The polycentricity index is calculated as follows:(1)$$I=-\sum_{1}^{N}\frac{Z_{i}\ln(Z_{i})}{\ln N}\cdot N\cdot R_{c}$$
where *N* is the number of sub-centers, which should meet three criteria: (1) the sub-center’s density is higher than the city density, (2) the adjacent polygons above the threshold are grouped together, and (3) the sub-centers should contain $\frac{10}{\sqrt{population}}\%$ or more of the city population. $Z_{i}$ is the proportion of the population living in the sub-center i to the population living in all the sub-centers. $R_{c}$ is the proportion of the population living in all the sub-centers to the total population size. The statistical descriptions of traffic congestion and the BE variables are presented in Table 3.

### 3.2. Variables

In this study, the dependent variable is one-way commute duration, which is affected by the independent variables and mediating variables simultaneously. The independent variables include individual socio-economic variables (excluding car ownership), the city-level built environment variables, and the neighborhood-level variables. The independent variables not only affect commute durations directly, but also affect commute durations via the mediating variables. Car ownership, traffic congestion, and commuting modes serve as mediating variables in this study. The mediating variables are affected by the independent variables, and, in turn, they affect the dependent variable.

### 3.3. Methods

The structural equation model (SEM) is a powerful method to model the complex relationships among exogenous variables and endogenous variables [31,55]. In addition, the SEM can provide the direct, indirect, and total effects of exogenous variables. It helps to better understand the inherent mechanism among the BE, traffic congestion, socio-economics, car ownership, commuting modes, and commuting durations. In this study, the mediating effects of traffic congestion, car ownership, and commuting modes are considered. The mediating effects of car ownership are largely attributed to the hierarchical decision theory [18,58], in which car ownership is affected by the built environment, and, in turn, affects travel behavior. The effects of the built environment on traffic congestion have been well-studied by existing studies [59]. Meanwhile, the effects of traffic congestion on commuting durations are also are observed in the literature [60]. Similarly, commuting modes are also associated with the built environment and commuting durations simultaneously. Thus, commuting modes also serve as a mediating variable in this study [61]. However, all the endogenous variables are assumed to be continuous in a traditional SEM [62]. Moreover, a traditional SEM cannot address unobserved heterogeneities.

In this study, a conceptual framework is proposed (Figure 1), in which three non-continuous endogenous variables are included: traffic congestion (binary variable), car ownership (binary variable), and commuting modes (categorical variable). Moreover, it is necessary to consider the unobserved heterogeneities at both the neighborhood and city levels, owing to the grouped nature of the data. Thus, a generalized multilevel structural equation model (GMSEM) is used to explore the impacts of the BE at two levels on commuting durations in this study. The GMSEM allows for the endogenous variables to be variant types of responses (e.g., binary variable and categorical variable) and can accommodate unobserved heterogeneities at two levels [6,62,63].

To examine whether the impacts of the BE variables at two levels vary between the male and female members of the same household couple, the multiple-group modeling approach is incorporated into the GMSEM. The multiple-group GMSEM can perform one single analysis, in which the parameters are estimated for both males (group 1) and females (group 2). In this study, a constrained model is first used to examine the relationships based on the total sample, in which all the parameters are assumed to be the same in the two groups. Then, an unconstrained model is applied to examine the difference between the two groups.

The Akaike information criterion (AIC) and Bayesian information criterion (BIC) are applied to confirm the effectiveness of the multi-group GMSEM. Smaller values of the AIC and BIC indicate better fitness. The AIC and BIC can be obtained based on Equations (2) and (3).
(2)AIC=−2ln(L)+2p
(3)BIC=−2ln(L)+pln(n)
where L represents the likelihood function, p represents the number of parameters, and n represents the number of the sample.

## 4. Results

The model fitness of the constrained and unconstrained models is shown in Table 4. The unconstrained model has smaller values of the AIC and BIC, indicating that it is a better fitting model. In other words, the unconstrained model performs better in fitting the complex relationships among two levels of the BE, traffic congestion, car ownership, commuting modes, and commuting durations.

### 4.1. Impacts of Multilevel BE on Mediating Variables

Traffic congestion and car ownership are the endogenous variables at the city and household levels, and should not be affected by gender. Thus, we assume that the impacts of the independent variables on traffic congestion and car ownership remain the same for the male and female members of the same household couple. The estimated results of the GMSEM models are shown in Table 5.

Among the city-level BE variables, the population density, polycentricity, and road areas per person show significant effects on traffic congestion. The population density shows a significantly positive impact on traffic congestion, whereas the polycentricity and road areas per person show significantly negative impacts.

Most of the variables show significant impacts on car ownership. Among the city-level BE variables, the population density, land use mix, polycentricity, public transit supply, road areas per person, and bus coverage rate show significantly negative impacts on car ownership. Regarding the neighborhood-level BE variables, the population density, distance to transit stations, and distance to the CBD show significantly negative impacts on car ownership, which are in line with most existing studies [31,64,65,66]. The facility accessibility shows a significantly positive impact on car ownership, which is inconsistent with most existing studies [67]. A total of two potential reasons may explain this unexpected result. First, diversified land use tends to provide more parking lots, which may encourage car ownership [68]. Second, although many studies believe that diversified land use reduces motorized travel demand, some researchers have found that diversified land use might stimulate car travel demand by reducing travel costs [69]. The impacts of the socio-economic variables indicate that older, well-educated people from households with more members and higher incomes tend to have a higher probability of owning a car.

The results demonstrate that the relationships between the two levels of BE variables and commuting mode choices differ between the male and female members of the same household couple. In addition, the BE variables are more influential to males’ commuting mode choices. Among the city-level BE variables, the population density and public transit supply show significantly positive impacts on commuting by active modes for males, whereas their impacts are not significant for females. The variable of road areas per person only shows a significantly negative impact on commuting by active modes for males. All the neighborhood-level BE variables show significant impacts on commuting by active modes for males, whereas the impact of the distance to the local CBD is not significant for females. The facility accessibility shows a significant impact on commuting by public transit for males, whereas its impact is not significant for females. Except for the above variables, the other BE variables show no different impacts for males and females. The polycentricity and land use mix show significantly positive impacts on active modes, suggesting that people living in a region with polycentricity and more balanced land use have a higher probability of commuting by active modes. The polycentricity, metro, road areas per person, and bus coverage rate show significant impacts on commuting by active modes. The population density at the neighborhood level has a significantly positive impact on commuting by active modes for both males and females. The distance to transit stations shows significantly positive impacts on commuting by active modes and public transit for both males and females.

### 4.2. Gender Differences in Impacts of Multilevel BE

The gendered relationships between the factors and commute durations are shown in Table 6. The findings demonstrate that the statistical significances of both the BE and socio-economic variables differ between the male and female members of the same household couple. Moreover, the total effects of the BE and socio-economic variables are weakened and strengthened via their corresponding indirect effects.

Among the city-level variables, traffic congestion shows a significant impact on commuting durations, suggesting that people living in more congested cities tend to have longer commuting durations. In terms of the total effects, most of the city-level BE variables show significant impacts on commuting durations, and males’ commute duration is more strongly associated with the BE variables. Specifically, the impacts of the population density and metro are significantly positive for both males and females. The direct effect of the population density is strengthened by the corresponding indirect effects, whereas the total effect of the population density is weakened by the corresponding indirect effects. The land use mix and polycentricity show significantly negative impacts on commuting durations for both males and females. The negative impact of the land use mix indicates that diversified land-use developments can reduce commute durations. This result is consistent with several previous studies [5,12]. The impact of the polycentricity is significantly negative, suggesting that polycentric developments reduce commute durations. This result may be explained by the fact that polycentric developments can moderate the commuting burdens caused by residential suburbanization, as people in polycentric cities have a higher probability of living and working around the same sub-centers. Additionally, the corresponding indirect effects strengthen the direct and negative effects of the land use mix and polycentricity. The impacts of public transit supply and road areas per person differ between the male and female members of the same household couple. The impacts of the population density at the two levels are significant for males, whereas its impacts are insignificant for females. Similar to the population density, the impact of the public transit supply is only significantly positive for males. The positive impacts of the metro and public transit supply are unexpected. These impacts may be explained by two reasons. First, first- and last-mile problems can take up a large proportion of commuting durations, especially in areas with low public transport accessibility. Second, the impact reflects the endogenous relationship among public transit services, commuting durations, and unobserved factors. For example, the improvement of public transit services may actually result from larger populations or city sizes, which may increase commuting burdens.

Regarding the neighborhood-level variables, most of the BE variables show significant impacts on commuting durations. Additionally, the impacts of the population density and facility diversity differ between the male and female members of the same household couple. Specifically, the distance to transit stations and the distance to the CBD show significantly positive impacts on commuting durations for both females and males, and the corresponding indirect effects weaken their direct effects. In contrast, the impacts of the population density and facility diversity on commuting durations are significant for males, whereas their impacts are insignificant for females. In addition, the corresponding indirect effects strengthen the direct effects of the population density and facility diversity.

As for factors at the individual level, the impacts of age, education, hukou, and electric bike ownership show the same statistical significance for both females and males. The positive impacts of education and hukou suggest that well-educated people with a local hukou tend to have longer commutes. Some variables produce different impacts for the male and female members of a household couple. Income and household size only show significantly negative impacts on the commute durations for females. This result may be because females tend to have greater family responsibilities [24], and thus will spend more time on their domestic and caring roles once they have economic supports. Car ownership shows a significant impact for males, whereas its impact is not significant for females. This result is partially supported by Maat’s study [70], who found that those who had a longer commute tended to commute by car. In our sample, the average commute duration of males is 22.20 min, which is 4 min longer than that of females. Compared with car commuters, active commuters have shorter commutes, while those commuting by public transit tend to have longer commute durations.

## 5. Discussion and Policy Implications

### 5.1. Discussion

This study has significant implications for whether and how the BE variables at different spatial levels affect commute durations. It enriches prior studies by including both neighborhood- and city-level BEs. Additionally, the findings highlight the importance of identifying the gendered relationships between BEs and commute durations. This section presents the key findings of this study.

First, the findings appear as additional evidence for BEs and commute durations by including city-level BE variables. The findings suggest that the BE variables at the two levels play critical roles in influencing commute durations. This result is similar to previous studies that found that the BE variables at different spatial levels are the key factors of travel-related issues [4,6]. Specifically, six factors at the city level (i.e., population density, land use mix, polycentricity, metro, public transit supply, and road areas per person) have significant impacts on males’ or females’ commuting durations, and all the BE variables at the neighborhood level are significant factors. The remarkable role played by the city-level BE variables may be explained by the fact that people’s workplaces are usually beyond their local areas, owing to the job–housing imbalance in China [35]. Therefore, to reduce commute durations, the findings in this study call for planning efforts at both the neighborhood and city levels.

Second, this study further demonstrates the gender differences in the impacts of the BE variables on commute durations. The results suggest that the BE variables have a greater impact on commute durations for males. Specifically, most of the BE variables show significant impacts for males, whereas only the impacts of the city population density, metro, land use mix, polycentricity, distance to transit stations, and distance to the CBD are statistically significant for females. This result suggests that males are more sensitive to BEs than females, and this difference is consistent with some existing studies that have found that males tend to attach more importance to their external environment, including their social and physical environments [26,71]. This result may be explained by the reason that females tend to take more family responsibilities, and thus their travel behavior is usually constrained. It is necessary for urban planners and transport practitioners to better understand the differences between the male and female members of household couples when developing targeted strategies to reduce commuting durations.

Third, the impacts of the population density at the city and neighborhood levels are positive. This result is inconsistent with most previous studies [8,12,58], in which researchers usually find that promoting high-density development strategies is important for reducing long commutes. However, some previous studies have provided findings similar to our results [10,72]. For example, Schwanen et al., found that the distance-shortening effect of population density can be eclipsed by the congestion effect [10]. This is reasonable, because higher population densities produce a greater mixed travel demand, composed of commuting and non-commuting trips, thus leading to traffic congestion during peak hours and longer commute durations [72]. The unexpected impacts of the population density in this study indicate that most Chinese cities with rather high population densities and severe traffic congestion would cause a prolonged commute duration if the population densities were to increase constantly. Therefore, promoting high-density development strategies may not be a solution to reduce commute durations in the context of Chinese cities.

Finally, we find that the impact of car ownership is significantly positive for males. This result suggests that male commuters who own a car tend to commute for a longer duration. Regarding the link between commuting and car ownership, prior studies have provided mixed evidence [12,31,73]. Prior studies from Western countries have found that car ownership is negatively associated with commute durations and distance [65,74], whereas some recent studies from China have found positive links between commuting distances and car ownership [12,75]. This difference may be partially explained by the extremely severe traffic congestion in Chinese megacities [76]. In such a context, commuting by car may not reduce commute durations. Thus, the improvements of public transit services are important for low-carbon transport initiatives, because they not only meet the motorized travel demand of long trips, but also mitigate traffic congestion compared with private cars.

### 5.2. Policy Implications

These findings provide policy implications for reducing commuting durations. First, policy makers should pay more attention to the impacts of the city-level BE variables on commute durations and try to mitigate the negative effects that are induced by overly compact development patterns. Specifically, maintaining a reasonable city population density and promoting polycentric development patterns help to shorten commute durations. Second, the urbanization process has caused large-scale rural-to-urban migrations and urban sprawl in China. Most migrant workers live in urban villages because they cannot afford the high rent in other areas. They have a strong travel demand for work and other activities, but they tend to have no car. In other words, migrant workers living in urban villages should have a higher motorized demand that needs to be satisfied by public transit systems. Moreover, the results suggest that people living farther from transit stations tend to have a longer commuting duration. Thus, it is important to increase the investments into developing public transit systems around urban villages, in order to reduce the imbalance of the public transport between the migrant workers and local residents. Finally, the results suggest that policy makers should pay more attention to reducing females’ commuting durations. Females tend to have greater family responsibilities [77], and thus their spatial flexibility is relatively lower. This can be partially confirmed by the greater impacts of the BE on males’ commuting behavior. Therefore, to design gender-equal transportation systems, urban planners and policy makers should pay more attention to females’ needs, such as preferences, safety, and comfort.

## 6. Conclusions

Using survey data from 3209 household couples across 329 communities nested within 97 cities, this study reveals the impacts of neighborhood- and city-level BE variables on commute durations, and identifies the gender gaps in these impacts. The mediating effects of traffic congestion, car ownership, and commuting modes are considered. The findings show that most of the BE variables at the two levels are key factors of commute durations. Moreover, the gender differences in these impacts are observed through the impact of the BE variables. Specifically, the population density, land use mix, polycentricity, metro, public transit supply, and road areas per person at the city level show significant impacts on commute durations for males, whereas only the population density, land use mix, polycentricity and metro show significant impacts for females. Regarding neighborhood-level BEs, the impacts of the population density, facility diversity, distance to transit stations, and distance to the CBD are statistically significant for males, whereas only the distance to transit stations and distance to the CBD show significant impacts for females. The direct effects of the BE variables are strengthened or weakened by the corresponding indirect effects via traffic congestion, car ownership, and commuting modes. These findings highlight the importance of understanding gender differences when designing gender-equal transportation systems.

Several limitations of this study should be noted in future studies. First, the causal links between the BE and commute durations are not addressed in this study, because the survey data are cross-sectional. Future studies need more longitudinal data to capture causal links. Second, only household couples are considered, and the commuting behaviors of other household members are ignored in this study. Other members’ travel behaviors may be associated with household couples’ commuting durations, which should be paid attention to in future studies. Third, the self-selection effect is not fully considered in this paper, because the attitudes and preferences for travel and land use are not surveyed in the travel survey.

## Figures and Tables

**Figure 1 ijerph-20-04851-f001:**
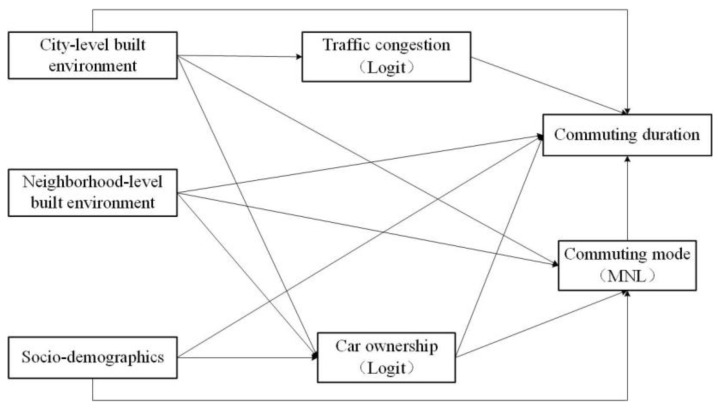
Conceptual framework.

**Table 1 ijerph-20-04851-t001:** Summary of causes of transport inequity in China.

Factor	Study	Method	Conclusion
Urban structure	[36]	Structural equation model	High property prices around city centers force low-income workers to move to suburban areas, where amenities are lacking.
	[37]	Multinomial logistic regression	Higher land use mixture forces low-income residents to leave the community, leading to poor access to services owing to its effect on housing rent.
	[38]	Geographical analysis	Affordable housing communities have poorer access to amenities.
	[39]	Analytic hierarchy process	Low-income workers usually live in suburbs, where transport options are limited.
Access to cars	[40]	Job accessibility model	Middle and low-income groups living in affordable housing mainly use public transit, whereas high-income groups mainly travel by private car.
	[41]	Multiple travel surveys, in-depth analysis	An increase in household incomes increases the gap in car ownership between the rich and poor.
	[42]	Logit and nested-logit models	Children from high-income households tend to go to school by car.
	[43]	Multinomial logistic model	Car ownership has an impact on the elderly’s healthcare-related travel behavior.
Alternative transport options	[44]	Questionnaire survey, in-depth interviews	Females, low-income migrant workers, and less-educated people have a lower probability of using the public bicycle sharing programs.
	[45]	Multivariate analysis	Low-income groups tend to have poorer access to transport options than others.
	[46]	Binary logistic regression	Dockless bike-sharing is more popular among high-income groups.
Public transport provision	[34]	Regression model	Poor access to public transport increases the commute duration of low-income workers.
	[47]	Index of dissimilarity and residential exposure	The uneven distribution of public transport services increases the residential segregation between local residents and migrants.
Institutional factors	[48]	Structural equation model	Danwei explains the difference in commuting behaviors among people.
	[49]	Case study	Disparities exist in housing options between people with local hukou and migrants.
	[50]	Regression model	Significant differences exist in commute duration among local residents, highly skilled migrants, and low-skilled migrants.
	[51]	Multiple linear regression	Inequalities exist in commuting burdens between local residents and low-skilled migrants.

**Table 2 ijerph-20-04851-t002:** Statistical descriptions of variables at the individual level.

Indicators	Description	Mean	Mean
		Female (*N* = 3209)	Male (*N* = 3209)
Individual socio-economics			
Age	Age in years	52.67	53.47
Education	0, otherwise; 1, own a college diploma or above	11.22	13.43
Household size	The number of household members	3.44	3.44
Hukou	0, otherwise; 1, local hukou	66.32	66.32
Income	Household yearly income (unit: 10,000 RMB)	6.20	6.20
Car ownership	0, otherwise; 1, own more than one car	19.41	19.41
Electric bike ownership	0, otherwise; 1, own more than one electric bike	57.03	57.03
Commuting attributes			
Commuting mode (reference: car)	1, active modes; 2, public transit; 3, others (motorcycle and electric bike)	1.46	0.39
Commuting duration	One-way commuting duration (unit: min)	18.25	22.20

**Table 3 ijerph-20-04851-t003:** Statistical descriptions of neighborhood- and city-level variables.

Indicators	Description	Mean	Standard Deviation
City-level variables			
Traffic congestion	0, otherwise; 1, the city entered the top 10 congested cities at least once in 2014	0.19	0.39
Population density	The proportion of population size and areas (unit: 10,000 persons/km^2^)	1.29	4.86
Land use mix	The degree of mixture of different POIs from different categories	0.67	0.09
Polycentricity	0, otherwise; 1, the city is polycentric	0.40	—
Metro	0, otherwise; 1, metro has been opened in the city	23.87	—
Public transit supply	The proportion of public transportation vehicles and population size (unit: /10,000 persons)	2.87	6.75
Road areas per person	The proportion of road areas and population size (unit: /10,000 persons)	19.04	13.34
Bus coverage rate	The areas covered by bus stops within a 500 m radius divided by the total areas suitable for setting public transport stops in the city	0.73	0.52
Neighborhood-level variables			
Population density	The proportion of population and neighborhood areas (unit: 10,000 persons/km^2^)	0.81	2.98
Facility diversity	Facility accessibility measurement with 7 types of facilities	3.71	1.59
Distance to transit station	The distance between home and the nearest transit station (unit: km)	2.10	1.28
Distance to local CBD	The distance between home and local CBD (unit: km)	4.90	7.08

Notes: POI represents point of interest; and CBD represents central business district.

**Table 4 ijerph-20-04851-t004:** Model fitness of the multi-group GMSEM.

Model Fit Measurements	Model-Based Values	
	Constrained	Unconstrained
AIC	17,669.78	15,744.28
BIC	18,123.37	16,651.46

**Table 5 ijerph-20-04851-t005:** Impact of multilevel BE on traffic congestion, car ownership, and commuting modes.

Variables	Constrained Part				Unconstrained Part											
	Male and Female				Male						Female					
	Traffic Congestion		Car Ownership		Commuting Mode (Reference: Car)						Commuting Mode (Reference: Car)					
	Direct Effect		Direct Effect		Total Effect (Active Modes)		Total Effect (Public Transit)		Total Effect (Others)		Total Effect (Active Modes)		Total Effect (Public Transit)		Total Effect (Others)	
	Coef.	S.E.	Coef.	S.E.	Coef.	S.E.	Coef.	S.E.	Coef.	S.E.	Coef.	S.E.	Coef.	S.E.	Coef.	S.E.
City-level variables																
Population density	0.089 **	0.011	−0.118 **	0.014	0.036 **	0.013	0.106	0.073	0.029 **	0.010	0.022	0.041	0.065	0.078	0.017 **	0.004
Land use mix	−0.012	0.008	−0.005 **	0.001	0.056 **	0.035	0.123	0.118	−0.114 **	0.022	0.118 **	0.047	0.149	0.135	−0.028 **	0.001
Polycentricity	−0.002 *	0.001	−0.017 *	0.010	0.165 **	0.038	0.124 **	0.041	0.018	0.014	0.101 **	0.021	0.105 **	0.036	−0.007	0.026
Metro	−0.103	0.088	0.028	0.108	−0.054	0.111	0.108 **	0.028	−0.008	0.006	−0.101	0.089	0.047 **	0.014	−0.004	0.004
Public transit supply	−0.042	0.038	−0.012 **	0.005	0.128 **	0.017	0.078 **	0.021	−0.016	0.035	0.070	0.071	0.109 **	0.025	−0.021	0.034
Road areas per person	−0.013 **	0.004	−0.005 **	0.001	−0.082 **	0.038	−0.069 **	0.028	0.045	0.051	−0.029	0.038	−0.076 **	0.033	0.038	0.051
Bus coverage rate	−0.008	0.027	−0.068 **	0.013	0.037	0.028	0.025 **	0.008	0.021	0.045	0.027	0.032	0.048 **	0.013	−0.015	0.087
Neighborhood-level variables																
Population density			−0.005 **	0.001	0.167 *	0.096	0.101	0.865	0.104 **	0.026	0.015 *	0.008	0.045	0.129	0.086 **	0.013
Facility accessibility			0.032 **	0.007	0.114 **	0.029	0.063 *	0.035	0.035 **	0.014	0.034 **	0.018	0.046	0.053	0.011 **	0.003
Distance to transit station			−0.002 *	0.001	0.022 **	0.011	0.196 **	0.063	0.003	0.005	0.023 **	0.010	0.069 **	0.031	0.004	0.007
Distance to local CBD			−0.003 **	0.001	−0.025 **	0.007	−0.020 **	0.009	−0.014	0.012	0.021	0.016	−0.023 *	0.012	−0.035	0.056
Individual socio-economics																
Age			0.005 **	0.001	0.019 **	0.004	−0.014 **	0.005	−0.024 **	0.002	0.009 **	0.004	−0.026 **	0.007	−0.015 **	0.007
Education			0.147 **	0.024	−0.138 **	0.036	−0.247 **	0.085	−0.023 **	0.007	−0.141 **	0.050	−0.189 **	0.024	−0.038 **	0.011
Household size			0.065 **	0.015	−0.018	0.032	−0.073	0.056	0.004	0.019	−0.013	0.060	−0.058 **	0.009	0.025	0.043
Hukou			0.086	0.121	−0.038	0.030	0.118 **	0.059	0.047	0.056	−0.025	0.034	0.147 **	0.045	0.031	0.025
Income			0.112 **	0.019	−0.015	0.041	−0.043	0.038	0.047	0.045	−0.018	0.054	−0.089	0.075	0.011	0.008
Electric bike ownership			0.007	0.046	0.114 **	0.024	0.045	0.42	0.054 **	0.016	0.049 **	0.021	0.078	0.105	0.017 **	0.003
Car ownership					−0.335 **	0.121	−0.247 **	0.062	−0.011 **	0.002	−0.198 **	0.057	−0.287	0.177	−0.104 **	0.027
Constant term	0.029 **	0.011	−0.038	0.048	0.234	0.307	−2.626 **	0.488	1.178 **	0.028	0.326	0.298	2.589 *	1.523	1.380 **	0.017

Notes: * *p* < 10%, and ** *p* < 5%.

**Table 6 ijerph-20-04851-t006:** Gender differences in the impact of multilevel BE on commuting durations.

Variables	Male						Female					
	Direct Effect		Indirect Effect		Total Effect		Direct Effect		Indirect Effect		Total Effect	
	Coef.	S.E.	Coef.	S.E.	Coef.	S.E.	Coef.	S.E.	Coef.	S.E.	Coef.	S.E.
City-level variables												
Traffic congestion	0.014 **	0.005					0.009 **	0.002				
Population density	0.013 *	0.067	−0.004 **	0.002	0.009 *	0.005	0.030 *	0.017	−0.003 *	0.002	0.028 *	0.016
Land use mix	−0.173 **	0.032	−0.018 **	0.005	−0.191 **	0.057	−0.093 **	0.030	−0.005 **	0.001	−0.098 **	0.032
Polycentricity	−0.114 **	0.028	−0.003 *	0.002	−0.117 **	0.028	−0.081 **	0.028	−0.001	0.001	−0.082 **	0.029
Metro	0.367 **	0.132	0.035	0.052	0.402 **	0.155	0.083 **	0.015	0.021	0.041	0.104 **	0.027
Public transit supply	0.081 **	0.032	0.003 *	0.002	0.084 **	0.031	0.112	0.127	0.002	0.008	0.114	0.138
Road areas per person	−0.100 **	0.032	−0.001 **	0.000	−0.101 **	0.033	−0.023	0.110	−0.001	0.005	−0.024	0.108
Bus coverage rate	−0.030	0.084	−0.015	0.020	−0.045	0.091	−0.063	0.070	−0.016	0.025	−0.079	0.077
Neighborhood-level variables												
Population density	0.082 **	0.030	0.006 **	0.002	0.088 **	0.031	0.015	0.052	0.003	0.010	0.018	0.057
Facility diversity	0.032 *	0.018	0.016 **	0.008	0.048 *	0.027	0.010	0.017	0.009	0.020	0.019	0.033
Distance to transit station	0.062 **	0.020	−0.002 **	0.001	0.060 **	0.021	0.106 **	0.035	−0.002	0.002	0.104 **	0.038
Distance to local CBD	0.020 **	0.006	0.003 **	0.001	0.023 **	0.005	0.060 **	0.011	−0.002 **	0.001	0.058 **	0.012
Individual socio-economics												
Age	−0.050	0.079	0.005	0.005	−0.045	0.082	−0.033	0.025	0.002	0.003	−0.031	0.024
Education	0.246 **	0.057	0.051 **	0.010	0.297 **	0.075	0.135 **	0.011	0.039 **	0.004	0.174 **	0.014
Household size	0.048 **	0.028	0.011 **	0.004	0.059 **	0.033	0.069	0.089	0.016	0.021	0.085	0.107
Hukou	0.043 **	0.012	0.019 **	0.008	0.062 **	0.021	0.193 **	0.032	0.012 **	0.004	0.205 **	0.049
Income	−0.020	0.098	−0.023	0.128	−0.043	0.117	−0.135 **	0.043	0.018 **	0.008	−0.117 *	0.041
Car ownership	0.210 **	0.100	0.017 **	0.006	0.227 *	0.106	0.130	0.087	0.005	0.096	0.135	0.091
Electric bike ownership	0.129	0.105	0.003	0.008	0.132	0.108	0.084	0.119	0.001	0.015	0.085	0.123
Commuting attributes												
Active modes					−0.102 **	0.022					−0.085 **	0.023
Public transit					0.381 **	0.102					0.221 **	0.105
Others					−0.009	0.011					−0.008	0.021
Constant term					2.254 **	0.621					3.028 **	1.023
Number of observations			3209				3209					

Notes: * *p* < 10%, and ** *p* < 5%.

## Data Availability

Not applicable.

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
