# Peer review of "Exploring the Relationships between Multilevel Built Environments and Commute Durations in Dual-Earner Households: Does Gender Matter?"

_ijerph, 2023, doi:10.3390/ijerph20064851_

Round 1

Reviewer 1 Report

This study does a nice job of demonstrating the impact multi-scale built environments on commuting time for dual-earner households. it is intriguing that it did a multigroup analysis to contrast the impact between the male and female groups. my only concern is whether all these differences in the coefficients of the two groups are significant as Table 6 did not show the. I'm hoping the authors will clarify it.

Author Response

Thanks for your insightful comments. As you suggest, identifying whether all these differences in the coefficients of the two groups is important in multi-group analysis. The GSEM is a new SEM model. So it cannot provide the statistical test of the differences in the coefficients of the two groups. However, we also disentangle the differences in the impacts between the male and female groups in Section 4.3, which is also the common analysis method of the differences between two groups.

Reviewer 2 Report

This paper presents new evidence on the multilevel built environment and commute duration in dual-earner households. Notably, it is a good study with a robust method and a large sample of data. Overall, this paper is well-organized and well-written. I only have several suggestions and comments for the authors.

First, the authors analyze the impacts of the multilevel built environment. I want to see the correlation matrix of all BE variables. Please include that in the response letter.

Second, related to the first comment, there may exist correlations between different levels of built environment variables. Also, explain why models 1, 2, and 3 are needed for the analysis, given that the assumptions of OLS have been violated. Have the multicollinearity issues been checked?

Third, self-selection effects are important in the links between the built environment and travel. However, the authors didn’t consider the self-selection effects. I suggest the authors should consider it or add it as a research limitation. 

Author Response

This paper presents new evidence on the multilevel built environment and commute duration in dual-earner households. Notably, it is a good study with a robust method and a large sample of data. Overall, this paper is well-organized and well-written. I only have several suggestions and comments for the authors.

[Response] Thanks for the comments. We have accordingly revised our manuscript, hoping that the revised manuscript can properly respond to all your comments.

Q2.1 First, the authors analyze the impacts of the multilevel built environment. I want to see the correlation matrix of all BE variables. Please include that in the response letter.

[Response] Thank you for your comments. The correlation matrix of all BE variables is shown as below.

City population density

Land use mix

Polycentricity

Metro

Public transit supply

Road areas per person

Bus coverage rate

Neighborhood population density

Facility diversity

Distance to transit station

Distance to local CBD

City population density

Land use mix

-0.038

Polycentricity

0.027

0.185

Metro

0.381

-0.274

0.036

Public transit supply

0.078

-0.143

0.219

0.541

Road areas per person

0.125

-0.082

0.079

0.337

0.559

Bus coverage rate

0.049

0.185

-0.012

0.090

0.116

0.211

Neighborhood population density

0.232

0.013

0.020

0.077

0.017

0.024

-0.017

Facility diversity

0.192

0.090

-0.120

0.178

0.176

0.233

0.115

0.022

Distance to transit station

-0.072

0.046

0.026

-0.152

-0.142

-0.174

0.013

-0.026

-0.150

Distance to local CBD

0.027

-0.010

-0.028

-0.080

-0.045

-0.108

0.023

-0.034

-0.098

0.223

Q2.2 Second, related to the first comment, there may exist correlations between different levels of built environment variables. Also, explain why models 1, 2, and 3 are needed for the analysis, given that the assumptions of OLS have been violated. Have the multicollinearity issues been checked?

[Response] Thank you for your comments. I think you mean why the constrained model is needed for the analysis. The constrained model is used as a comparison to confirm the effectiveness of the unconstrained model. Because we also want to know whether it is necessary for us conduct a multi-group analysis, we should compare it with the single-group analysis (constrain model). And the results actually confirm that the unconstrained model performs better than the constrained model.

We have tested the multicollinearity issues before estimating all models.

Q2.3 Third, self-selection effects are important in the links between the built environment and travel. However, the authors didn’t consider the self-selection effects. I suggest the authors should consider it or add it as a research limitation.

[Response] Thank you for your comments. As you suggest, we have it as a limitation (lines 515-517): Third, the self-selection effect is not fully considered in this paper because attitudes and preferences for travel and land use are not surveyed in the travel survey.

Reviewer 3 Report

This study investigated relationships between neighborhood- and city-level built environment (BE) and commute using a generalized multilevel structural equation model duration. We consider that the title of the paper is meaningful and that the abstract adequately summarizes the paper. The highlights and the keywords are also meaningful. Overall, the paper is suitable for "IJERPH". However, this paper requires better justification of the approaches used, as well as elaboration of the scientific innovation.

1.Introduction

In the second paragraph, the author claims that many western urban researchers have studied the relationship between BE and commute, but in the first paragraph, all the evidence is from China. It is suggested to supplement the first paragraph with evidence of increased commute duration in western cities. The authors should add evidence about commuting congestion in western cities to the first paragraph.

2. Literature review

ž   Title 2.4 is inaccurate. According to the author's statement, this part is about the gaps in the previous research and the necessity of this research.

3.1. Data sources

ž   We know that the definition of independent variable, dependent variable and mediator variable in structural equation model (SEM) is very clear. For readers who are less familiar with SEM, the current statement about variables is confusing. It is recommended that the author add a subsection to make a systematic statement on these three variables separately.

3.2. Methods

ž   Why did the author choose traffic congestion, car ownership and commuting mode as mediator variables? The reason should be provided.

ž   References need to be provided to justify the generalized multi-level structural equation model (GMSEM).

ž   The author should address the evaluation criteria of Akaike's information criterion (AIC) and Bayesian information criterion (BIC). When AIC and BIC are in which threshold range or quantification interval, can the fitting effect of the model be verified?

ž   The multi-collinearity diagnostics of the independent variables should be added to Tables 5, 6, otherwise the calibration results may be wrong.

5.1. Discussion

ž   In terms of the correlation between BE and commuting duration, the authors argue for city-level versus community-level variability as well as gender variability, which is the innovation of this study. It is suggested that the authors rationalize the reasons for these two discrepancies in the context of the rapid development of China's transportation, so as to provide clearer policy implications.

Author Response

This study investigated relationships between neighborhood- and city-level built environment (BE) and commute using a generalized multilevel structural equation model duration. We consider that the title of the paper is meaningful and that the abstract adequately summarizes the paper. The highlights and the keywords are also meaningful. Overall, the paper is suitable for "IJERPH". However, this paper requires better justification of the approaches used, as well as elaboration of the scientific innovation.

[Response] Thanks for the comments. We have accordingly revised our manuscript, hoping that the revised manuscript can properly respond to all your comments.

Q3.1 Introduction In the second paragraph, the author claims that many western urban researchers have studied the relationship between BE and commute, but in the first paragraph, all the evidence is from China. It is suggested to supplement the first paragraph with evidence of increased commute duration in western cities. The authors should add evidence about commuting congestion in western cities to the first paragraph.

[Response] Thank you for your comments. As you suggest, we have added evidence of commuting characteristics in western cities in the first paragraph as follows (lines 25-28): Almost all commuters in large cities over world are suffering from commuting bur-dens. For example, the number of inter-city commuters in Germany has increased from 2442630 in 2004 to 3123924 in 2014 [1], which has caused increased traffic loads and frequent congestion [2].

Q3.2 Literature review Title 2.4 is inaccurate. According to the author's statement, this part is about the gaps in the previous research and the necessity of this research.

[Response] Thank you for your comments. The title in last version is a mistake, we have revised it as follows in the new version: Comments on previous studies.

Q3.3 Data sources We know that the definition of independent variable, dependent variable and mediator variable in structural equation model (SEM) is very clear. For readers who are less familiar with SEM, the current statement about variables is confusing. It is recommended that the author add a subsection to make a systematic statement on these three variables separately.

[Response] Thank you for your comments. As you suggest, we have added a subsection to explain the variables in the study as follows (lines 237-245): 3.2. Variables

In this study, the dependent variable is one-way commute duration, which is affected by the independent variables and mediating variables simultaneously. The independent variables include individual socio-economic variables (excluding car ownership), city-level built environment variables and neighborhood-level variables. The independent variables not only affect commute duration directly, but also affect commute duration via the mediating variables. Car ownership, traffic congestion and commuting mode serve as mediating variables in this study. The mediating variables are affected by the independent variables, and in turn they affect the dependent variable.

Q3.4 Why did the author choose traffic congestion, car ownership and commuting mode as mediator variables? The reason should be provided.

[Response] Thank you for your comments. As you suggest, we have added some sentences to explain the mediating variables in the study as follows (lines 252-259): The mediating effects of car ownership is largely attributed to the hierarchical decision theory [18, 58], in which car ownership is affected by the built environment, and in turn affects travel behavior. The effects of built environment on traffic congestion have been well studied in existing studies [59]. Meanwhile, the effects of traffic con-gestion on commuting duration are also are observed in the literature [60]. Similarly, commuting mode is also associated with built environment and commuting duration simultaneously. Thus, commuting mode also serves as a mediating variable in this study [61].

Q3.5 References need to be provided to justify the generalized multi-level structural equation model (GMSEM).

[Response] Thank you for your comments. As you suggest, we have added the following references for the GSEM in the study.

Yin, C., J. Zhang, and C. Shao, Relationships of the multi-scale built environment with active commuting, body mass index, and life satisfaction in China: A GSEM-based analysis. Travel Behaviour and Society, 2020. 21: p. 69-78.

Zhang, L., Zhang, J., 2018. Perception of small tourism enterprises in Lao PDR re-garding social sustainability under the influence of social network. Tour. Manag. 69, 109–120.

Vu H T, Ha N M. A study on the relationship between diversification and firm performance using the GSEM method[J]. Emerging Markets Finance and Trade, 2021, 57(1): 85-107.

Q3.6 The author should address the evaluation criteria of Akaike's information criterion (AIC) and Bayesian information criterion (BIC). When AIC and BIC are in which threshold range or quantification interval, can the fitting effect of the model be verified?

[Response] Thank you for your comments. Because the values of AIC and BIC are directly related with the number of parameters, more parameters in the complex models will cause bigger values. Thus, there is no threshold range for two values. We can only compare the values of models for the same investigation. We are sorry for that we didn’t present clear explanation in last version about how to use the AIC and BIC. In the new version, we have supplied it as follows (lines 281-283): The Akaike's information criterion (AIC) and Bayesian information criterion (BIC) are applied to confirm the effectiveness of the multi-group GMSEM. Smaller values of AIC and BIC indicate better fitness.

Q3.7 The multi-collinearity diagnostics of the independent variables should be added to Tables 5, 6, otherwise the calibration results may be wrong.

[Response] Thank you for your comments. Before we conduct the GSEM analysis, we have tested the multi-collinearity diagnostics of the independent variables. All independent variables included in the study have pasted the multi-collinearity diagnostics. Because of the length and word limitation, we didn’t include it in the manuscript.

Q3.8 terms of the correlation between BE and commuting duration, the authors argue for city-level versus community-level variability as well as gender variability, which is the innovation of this study. It is suggested that the authors rationalize the reasons for these two discrepancies in the context of the rapid development of China's transportation, so as to provide clearer policy implications.

[Response] Thank you for your comments. Two main innovations of this study include: (1) considering both city-level and community-level BE; and (2) identify the gender differences in the relationship between BE and commuting duration.

For the first point, our focus is including both levels of BE variables, but not the city-level versus community-level variability. Because the measurements at two levels are not the same, it is a little difficult to clarify city-level versus community-level variability. However, your comments are insightful and we actually need to further discuss it in the context of the rapid development of China's transportation, so as to provide clearer policy implications. So in the new version, we have supplied more discussions about the results on both levels of BE (lines 421-432) and (lines 446-458): First, the findings appear as additional evidence for the BE and commute duration by including city-level BE variables. The findings suggest that the BE variables at the two levels play critical roles in influencing commute duration. This result is similar to previous studies that found that the BE variables at different spatial levels are key fac-tors of travel-related issues [1, 4]. Specifically, six factors at the city level (i.e., population density, land use mix, polycentricity, metro, public transit supply and road areas per person) have significant impacts on males' or females' commuting duration, and all BE variables at the neighborhood level are significant factors. The remarkable role played by city-level BE variables may be explained by the fact that people's workplaces are usually beyond local areas owing to the job-housing imbalance in China [35]. Therefore, to reduce commute duration, the findings in this study call for planning efforts at both the neighborhood and city levels.

Third, the impacts of population density at the city and neighborhood levels are positive. This result is inconsistent with most previous studies [8, 12, 58], in which re-searchers usually find that promoting high-density development strategies is important for reducing long commutes. However, some previous studies have provided findings similar to our results [10, 68]. For example, Schwanen et al. found that the distance-shortening effect of population density can be eclipsed by the congestion effect [10]. This is reasonable because higher population densities produce greater mixed travel demand composed of commuting and non-commuting trips, and thus leading to traffic congestion during peak hours and longer commute duration [68]. The unexpected impacts of population density in this study indicate that most Chinese cities with rather high population densities and severe traffic congestion would cause a pro-longed commute duration if population densities were to increase constantly. There-fore, promoting high-density development strategies may not be a solution to reduce commute duration in the context of Chinese cities.

Focusing on the first point, we also provide two policy implications as follows (lines 472-485): The findings provide policy implications for reducing commuting duration. First, policy makers should pay more attention to the impacts of the city-level BE variables on commute duration and try to mitigate the negative effects induced by overly compact development patterns. Specifically, maintaining a reasonable city population density and promoting polycentric development patterns help shorten commute duration. Second, the urbanization process has caused large-scale rural-to-urban migrations and urban sprawl in China. Most migrant workers live in urban villages because they cannot afford the high rent in other areas. They have strong travel demand for work and other activities, but they tend to have no car. In other word, migrant workers living in urban villages should have higher motorized demand that needs to be satisfied by public transit systems. Moreover, the results suggest that people living farther from transit stations tend to have longer commuting duration. Thus, it is important to increase investments in developing public transit systems around urban villages to reduce the imbalance in the public transport between the migrant workers and local residents.

For the second point, we have added potential explanations about the gender differences in the relationship between BE and commuting duration as follows (lines 438-442): This result suggests that males are more sensitive to BE than females, and this difference is consistent with some existing studies finding that males tend to attach more importance to external environment, including social and physical environments [26, 67]. This result may be explained by the reason that females tend to take more family responsibilities, and thus their travel behavior is constrained.

For the second point, we also provide some policy implications as follows (lines 485-491): Finally, the results suggest that policy makers should pay more attention to reducing females' commuting duration. Females tend to have greater family responsibilities [72], and thus their spatial flexibility is relatively lower. It can be partially confirmed by the greater impacts of BE on males' commuting behavior. Therefore, to design gender-equal transportation systems, urban planners and policy makers should pay more attention to females' needs, such as preferences, safety, and comfort.

Reviewer 4 Report

The paper - "Exploring the relationships between multilevel built environment and commute duration in dual-earner households: Does gender matter?" - is well written with appropriate introduction and motivation, literature review discussion with potential gaps in the prior studies, methodological layout, results, and the policy implications of the study. In particular, the author discusses the impact of the built environment on commute duration and how the impacts differ between male and female members of the same household couple. The results and findings do have policy implications for the design of gender-equal transportation systems. However, some clarification on the results is needed.

1. The study suggests that distance to transit station and distance CBD have negative impacts on car-ownership. This seems to be counterintuitive considering that those who live further away should be more dependent on personal modes of travel. Please explain.

2. Considering that commute is an induced travel demand for work, why would male be more sensitive to built environment than female?

Author Response

The paper - "Exploring the relationships between multilevel built environment and commute duration in dual-earner households: Does gender matter?" - is well written with appropriate introduction and motivation, literature review discussion with potential gaps in the prior studies, methodological layout, results, and the policy implications of the study. In particular, the author discusses the impact of the built environment on commute duration and how the impacts differ between male and female members of the same household couple. The results and findings do have policy implications for the design of gender-equal transportation systems. However, some clarification on the results is needed.

[Response] Thanks for the comments. We have accordingly revised our manuscript, hoping that the revised manuscript can properly respond to all your comments.

Q4.1 The study suggests that distance to transit station and distance CBD have negative impacts on car-ownership. This seems to be counterintuitive considering that those who live further away should be more dependent on personal modes of travel. Please explain.

[Response] Thank you for your insightful comments. As you suggest, it may be more reasonable that those who live further away should be more dependent on personal modes of travel. However, many residents still cannot afford a car in China (Jiang et al. 2017). Moreover, housing prices are high in areas near CBD and transit station in China. Thus, people living in these areas can afford a car more easily. Although those who live farther from transit station and CBD may not afford a car, they may have other choices such as electric bicycle, which is a common personal mode of travel in China.

Jiang Y, Gu P, Chen Y, et al. Influence of land use and street characteristics on car ownership and use: Evidence from Jinan, China[J]. Transportation Research Part D: Transport and Environment, 2017, 52: 518-534.

Q4.2 Considering that commute is an induced travel demand for work, why would male be more sensitive to built environment than female?

[Response] Thank you for your comments. As you suggest, we have supplied potential explanations in the manuscript as follows (lines 436-440): This result suggests that males are more sensitive to BE than females, and this difference is consistent with some existing studies finding that males tend to attach more importance to external environment, including social and physical environments [26, 71]. This result may be explained by the reason that females tend to take more family responsibilities, and thus their travel behavior is usually constrained.